# Computer tomography guided thoracoscopic resection of small pulmonary nodules in the hybrid theatre

Ioannis Karampinis[1,2☯], Nils Rathmann[3☯], Michael Kostrzewa[3], Steffen J. Diehl[3], Stefan O. Schoenberg[3], Peter Hohenberger[2], Eric D. Roessner[2,4]*

1 Division of Thoracic Surgery, The Royal Brompton and Harefield NHS Foundation Trust, London, United Kingdom, 2 Division of Surgical Oncology and Thoracic Surgery, University Medical Center Mannheim, Medical Faculty Mannheim, Heidelberg University, Mannheim, Germany, 3 Institute of Clinical Radiology and Nuclear Medicine, University Medical Center Mannheim, Medical Faculty Mannheim, Heidelberg University, Mannheim, Germany, 4 Academic Thoracic Center, University Medical Center Mainz, Johannes Gutenberg University Mainz, Germany

☯ These authors contributed equally to this work.
* eric.roessner@unimedizin-mainz.de

**Data Availability Statement:** All relevant data are within the manuscript and its Supporting Information files.

## Abstract

### Purpose

Thoracic surgeons are currently asked to resect smaller and deeper lesions which are difficult to detect thoracoscopically. The growing number of those lesions arises both from lung cancer screening programs and from follow-up of extrathoracic malignancies. This study analyzed the routine use of a CT-aided thoracoscopic approach to small pulmonary nodules in the hybrid theatre and the resulting changes in the treatment pathway.

### Methods

50 patients were retrospectively included. The clinical indication for histological diagnosis was suspected metastasis in 46 patients. Technically, the radiological distance between the periphery of the lesion and the visceral pleura had to exceed the maximum diameter of the lesion for the patient to be included. A spiral wire was placed using intraoperative CT-based laser navigation to guide the thoracoscopic resection.

### Results

The mean diameter of the lesions was 8.4 mm (SD 4.27 mm). 29.4 minutes (SD 28.5) were required on average for the wire placement and 42.3 minutes (SD 20.1) for the resection of the lesion. Histopathology confirmed the expected diagnosis in 30 of 52 lesions. In the remaining 22 lesions, 9 cases of primary lung cancer were detected while 12 patients showed a benign disease.

### Conclusion

Computer tomography assisted thoracoscopic surgery (CATS) enabled successful resection in all cases with minimal morbidity. The histological diagnosis led to a treatment change

**Funding:** No funding was received for this study.

**Competing interests:** IK has no competing interests to declare. NR has received travel grands from Siemens Healthcare GmbH. The ICRN has research agreements with Siemens Healthcare GmbH. MK has no competing interests to declare. SJD has no competing interests to declare. The ICRN has research agreements with Siemens Healthcare GmbH. SOS has no competing interests to declare. The ICRN has research agreements with Siemens Healthcare GmbH. PH has no competing interests to declare. EDR has received travel grands from Siemens Healthcare GmbH. This does not alter our adherence to PLOS ONE policies on sharing data and materials.

in 42% of the patients. The hybrid-CATS technique provides good access to deeply located small pulmonary nodules and could be particularly valuable in the emerging setting of lung cancer screening.

## Introduction

The introduction of lung cancer screening and the implementation of structured follow-up programs for solid malignancies have led to a rapid increase in the number of small pulmonary nodules detected in CT scans [1, 2]. Parallel to that, the spread of the thoracoscopic surgery has given the medical community access to pulmonary lesions avoiding the morbidity associated with the thoracotomy [3].

Both the lung cancer screening programs and the guidelines of the Fleischner Society focus on incidentally detected pulmonary nodules and less on lesions diagnosed during follow-up CT scans in cancer patients [4]. In oncological patients, more than two lesions occurring newly in the lung are considered to represent metastases. This scenario requires a different treatment approach depending on the primary tumor and on whether a potentially curative or palliative status is assumed. Malignancies such as colorectal cancer are considered curable in an oligometastatic stage and resection of colorectal pulmonary metastases is routinely performed worldwide [5]. Interestingly enough, the most common type of surgery performed in thoracic centres is pulmonary metastasectomy representing almost 50% of the work load of a European thoracic center [6, 7]. However, a recent analysis of the "National Lung Cancer Screening Trial" showed an increased risk of interventions following false positive diagnosis [8]. This underlines the dilemma on how to treat a patient with a cured primary tumor who is referred with an unclear pulmonary lesion, especially if previous CT scans are not available for comparison [9].

Computed tomography assisted thoracoscopic surgery (CATS) was introduced to enable access and achieve histological diagnosis from small pulmonary nodules, located deep in the lung parenchyma. The aim of this study was to examine the histology of pulmonary nodules resected with CATS and the influence of the histology on the treatment algorithm.

## Materials and methods

The study was approved by the local ethics committee (Ethikkommission II, Medical Faculty Mannheim, University of Heidelberg, #2016-865R-MA). Due to the retrospective nature of the study an informed consent was not required.

Patients who had undergone CATS from June 2014 to March 2019 were included in this retrospective analysis. The following inclusion criteria were applied in order to select patients for the CATS procedure:

- Patients with a maximum of 2 pulmonary nodules

- Either suspected pulmonary metastasis in patients with previous malignancy (46 cases) or a persistent ground glass opacity (GGO) without solid components (simple GGO)

- Indication for surgical biopsy confirmed in the multidisciplinary team meeting

- Distance between the periphery of the lesion and the visceral pleura exceeded the maximum diameter of the lesion (lesion depth/lesion diameter ratio > 1).

Patients with a depth/diameter ratio < 1 were treated by VATS which was performed in 131 patients during the same time period. Patients with ground glass opacities without solid components in the lung tissue were included in the study regardless of the depth of the lesion, due to the fact that these lesions can hardly be detected through thoracoscopic palpation [10]. All 50 patients were treated by the same surgical-radiological team. Patients with lesions fulfilling the above mentioned criteria but non suitable for wire marking (e.g. due to proximity to major vessels or surrounded by major vessels) were excluded from this analysis.

## CT guided wire placement

The procedure was conducted under general anaesthesia. After intubation and ventilation the patient was turned to a lateral decubitus position and the surgical field was prepared for the thoracoscopic access. The C-arm cone beam CT (CBCT) was then performed according to the protocol published earlier [10]. Using the inherent laser navigation system of the multiaxis C-arm system (Syngo X-Workplace; Siemens Healthcare GmbH, Germany) an 18-gauge marking wire with a spiral end (Somatex Lung Marker; Somatex Medical Technologies GmbH, Germany) was positioned. After lesion-marking a repeat CBCT scan was performed to verify the correct position of the wire and detect potential complications.

## Thoracoscopic resection

The lesion was resected through a standard 3-port thoracoscopic approach (Copenhagen approach). The previously placed wire was shortened in order to allow thoracoscopic manipulation. Intraoperatively, the marked part of the lung was lifted up by holding the wire. After palpation of the wire tip, the wedge was resected using a commercially available stapling device. A 20 Ch. thoracic drain was routinely inserted in all but 2 cases. The complete workflow has been published before [10].

## Results

50 patients (22 males, 28 females; mean age 63.1 years, SD 10.6) with a total of 52 lesions were included to this study. The histopathological findings of the resected specimens are summarized on Table 1. For detailed characteristics of the patients, the histopathology and the procedure see the online supplementary material (Tables 2 and 3 in S1 File). The mean diameter of the lesions resected (for the calculation the longest diameter of the lesion was measured in the pre-interventional CT scan) was 8.41 mm (SD 4.27 mm). The mean lesion depth measured in the pre-interventional CT scan was 18.3 mm (SD 10.3). The mean depth to diameter ratio was

**Table 1. Histopathological findings (n = 50 patients).**

| n | Suspected diagnosis | Final diagnosis | Treatment change |
|---|---|---|---|
| **29** | Metastatic cancer | Metastatic cancer | Further oncological treatment |
| **6** | Metastatic cancer | Primary lung cancer | Lobectomy n = 4 |
| | | | Follow-up n = 2 |
| **11** | Metastatic cancer | Benign disease | No treatment change n = 9 |
| | | | Antitubercular treatment N = 2 |
| **3** | GGO | Primary lung cancer | Lobectomy n = 2 |
| | | | Follow-up n = 1 |
| **1** | GGO | Benign disease | Antitubercular treatment |

n: patient number, GGO: ground glass opacity.

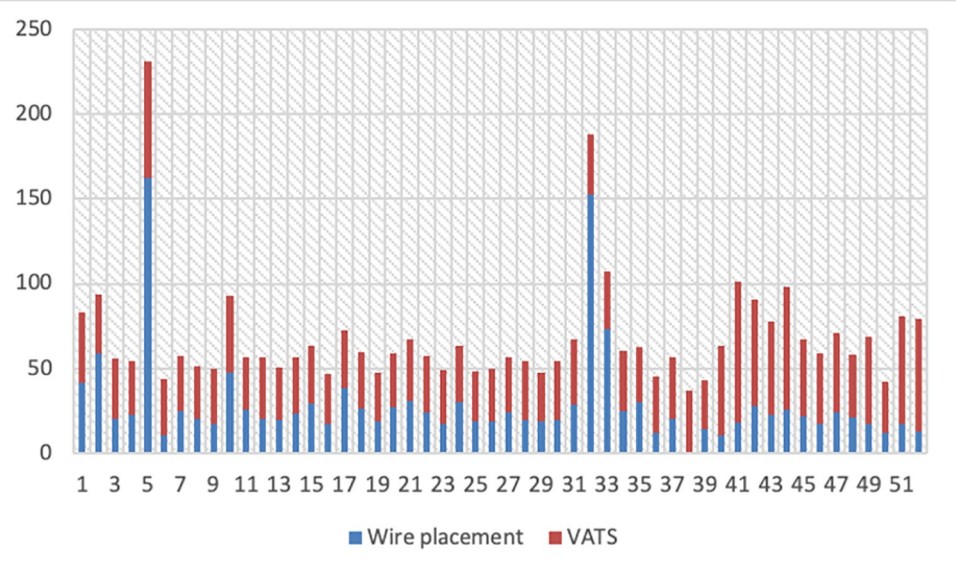

**Fig 1. Time (min) for guide wire placement and thoracoscopic resection per patient.**

2.45 (SD 1.38). Lesions with a depth to diameter ratio under 1 were only GGOs without solid components.

In two patients resection of two lesions was performed during one procedure (#14 and 28). One patient (#29) was operated two separate times for one lesion in each lung. The average hospital stay was 4.7 days (SD 2.8 range 2–20). Three patients stayed significantly longer in the hospital than the average. In one patient a prolonged air leak was treated conservatively (#5), while another patient required surgical reintervention (VATS) for a postoperative hae-mothorax (#11). In another patient, a complex anticoagulation algorithm had to be followed (#43). Two cases had to be converted to thoracotomy due to a system failure (collision of the CT-arm with the operating table after the installation of a new update).

The mean time required for wire placement and confirmation of the correct position of the wire was 29.4 minutes (SD 28.48, Fig 1) and for the thoracoscopic resection was 42.3 minutes (SD 20.1). There were two cases exceeding 2 hours of intervention time. In the first case colli-sion between the C-arm and the operating table was experienced after installation of a system update as described above. In the second case, the patient developed a tension pneumothorax after the placement of the wire and a thoracic drainage had to be placed before continuing the procedure. Overall, the rate of grade $\geq$ 3 complication according to the Clavien-Dindo classifi-cation after the CATS procedure was 2% [11].

## Analysis of the histological reports

A comparison of the histological reports of the lung tissue with the histology of the primary tumor revealed that 75% (n = 39) of the 52 resected lesions were malignant. However, in 37% (n = 19) of patients with previous malignant tumours, the histological report of the lung nod-ule did not match that of the primary. In 25% (n = 13) of the patients the malignancy could not be confirmed and a benign nodule was diagnosed (Fig 2).

When analysing the nine patients with a previous history of primary lung cancer or current ground glass opacity, the histological report revealed an adenocarcinoma of the lung in 7 patients, one case each of tuberculosis and benign lesion. Out of 11 patients with

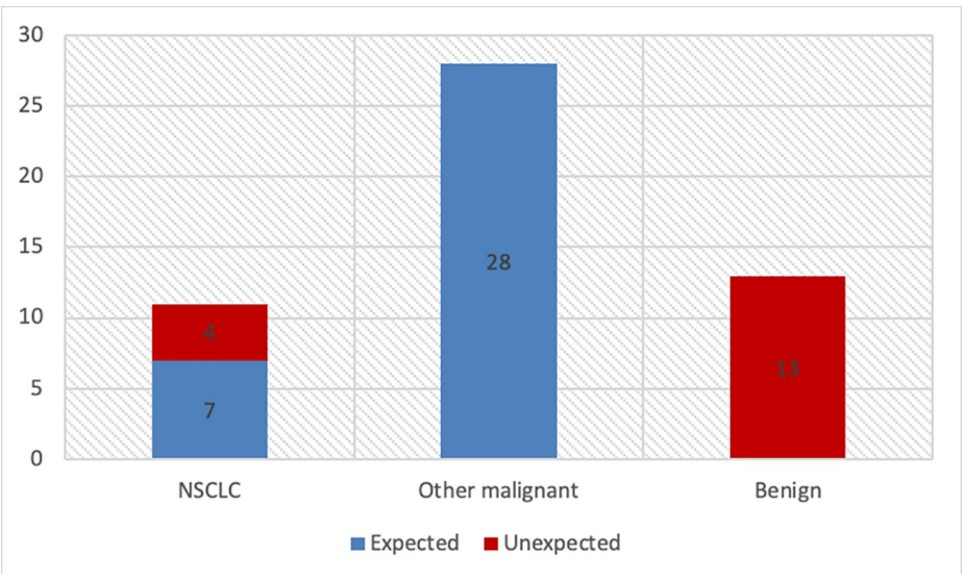

**Fig 2. Distribution of histological results.**

gastrointestinal primary tumors, five patients had corresponding lung metastases, one patient had an adenocarcinoma of the lung, 1 patient showed a tuberculotic affection (Fig 3) and the other patients had 4 benign lesions like intrapulmonary lymph nodes.

Of the 13 lesions with a benign histology, 4 lesions were of inflammatory origin such as tuberculosis (n = 3), 4 lesions showed a benign lymphnode, 1 lesion an atelectasis, 1 lesion a granuloma and 2 lesions a hamartoma, respectively.

## Discussion

Due to the high imaging quality and the low radiation exposure, the indication for low-dose CT scans has expanded both to the screening of primary lung cancer and to the follow-up of other malignancies. This will certainly lead to a further increase in the detection rate of small, nonspecific pulmonary lesions. The main problem resulting is the potential overtreatment

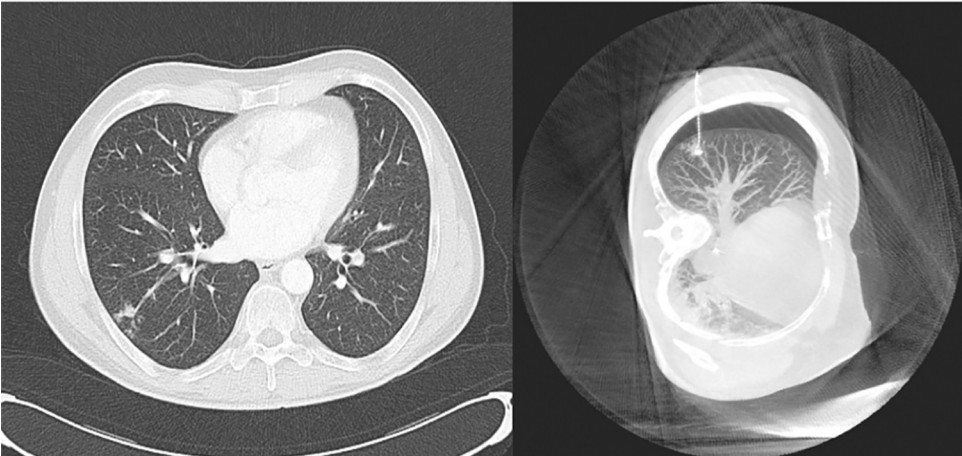

**Fig 3. Suspicious right lower lobe lesion which turned out to be tuberculosis.**

caused by the invasive procedures performed due to falsely positive findings on chest CT scans as the NLST trial confirmed [8]. On the other hand, resecting lesions that prove to be metastases is probably beneficial for the patient, since improved survival has been shown following resection of pulmonary metastases. Furthermore, the current trend in the treatment of oligometastatic tumors is towards more aggressive surgical approaches. As a consequence, it is essential to find the optimal balance between oncological benefit and overtreatment of patients with small pulmonary nodules.

Several methods have been developed to support the intraoperative localization of small lesions prior to thoracoscopic resection [12–14]. The most profound approaches are preoperative guide wire placement by the interventional radiologists and electromagnetically guided bronchoscopy [15]. The preoperative CT-guided wire placement has the main advantage of being more easily accessible in various institutions. However, the dislocation rate of the wire goes up to 33%, particularly on the way from the radiology department to the operating theatre [16]. Development of an undetected pneumothorax is reported in another 12% which might make it dangerous for the patients if the time interval from wire placement to resection by VATS is long [17]. Furthermore, the patient discomfort and stress as well as the burden associated with the organisation of both procedures is not negligible.

Electromagnetic bronchoscopy is another modern method providing access to small peripheral lesions. The NAVIGATE study, a prospective multicentre study about electromagnetic bronchoscopy (ENB) lesion marking for thoracoscopy reported success rates up to 94% in experienced hands and a median procedural time of 25 minutes [18]. However, the risk of adverse event following ENB was not negligible (4.3% risk of pneumothorax and 1.5% risk of relevant haemorrhage, [18]). Furthermore, the procedure complexity and the learning curve for achieving proficiency in accessing the peripheral lesions with an acceptable diagnostic rate is significant, even in highly specialised centres [19]. Several methods for preoperative staining of the lesion with dyes like indocyanine green with either using a percutaneous needle, electromagnetic bronchoscope or even after intravenous injection have also been described [20, 21].

The intraoperative guide wire marking with subsequent thoracoscopic resection (described as iVATS by Ng CSH [22] or CATS by our group [10]) was developed in order to combine the advantages of the CT-guided wire placement with the benefits of the immediate resection via thoracoscopy. This approach combines the advantages of the CT-guided wire placement avoiding the problems of dislocation, clinically relevant pneumothorax, patient discomfort or the organisational burden. Furthermore, the spiral guide wire used, allows the intraoperative palpation of the peripheral margin of the lesion which is extremely helpful in defining the resection margins and preventing incomplete resections. The method is easy to learn as it is very similar to the traditional thoracoscopy and the learning curve is therefore rather flat. In our patient cohort we noticed an increase in the time required for thoracoscopic resection in the last 10 cases, which can probably be explained by the integration of new members to the team and the associated learning phase.

The success rate of direct CT-guided lesion marking was 80% (in 20% of the cases the lesion was not directly penetrated by the wire but the positioning of the wire enabled the localisation of the lesion) with an intraoperative pneumothorax rate of 82%. These results confirm the findings of Ng CSH, who successfully managed to localize tiny indeterminate pulmonary nodules in a group of 32 patients [22]. The authors described a pneumothorax rate of 47% in the post-interventional CT scan, which is significantly lower than the rate observed in our study. This increased post-interventional pneumothorax rate could be potentially explained by the depth of the resected lesions, which, in our study, was almost 1 cm deeper. The pneumothorax was clinically irrelevant in 49/50 patients since single lung ventilation with consequent thoracoscopy could be established immediately after the wire marking.

The histological results of 52 lesions resected with CATS were analyzed in order to detect if the histopathology from the resected pulmonary lesions corresponded with the histology of the primary tumors. In 37% of the cases the histology of the lung lesion differed from the one of the primary changing the therapeutic algorithm. Furthermore, 9 cases of primary lung cancer were discovered in an early stage allowing for further oncological treatment with curative intent. Furthermore, 13 lesions could be classified as benign despite initially suspected malignancy, and the patients could be relieved from the stage of suffering from metastatic disease. Three patients were diagnosed with tuberculosis and were treated accordingly.

The results of this study confirm the high detection rate of lung cancer and other malignancies by CT scans performed for oncologic follow-up or screening purposes. The CATS procedure is associated with very low morbidity, early recovery and the advantage of simultaneous diagnosis and treatment without the burden of coordinating surgery with radiological guide wire placement, wire dislocation and without the morbidity of iatrogenic pneumothorax.

## Limitations

The collection of the data in this study was performed prospectively but the interpretation and process of the information was performed retrospectively, encompassing a certain possibility of selection bias. Furthermore, although this is a large cohort of patients with very small pulmonary nodules, the patients' characteristics regarding the primary tumors are heterogeneous, which limits the generalizability of the conclusions.

## Conclusion

In this study 40% of the patients underwent surgery for a lesion that was proven to be histologically different from what the MDT was expecting, including 9 new cases of primary lung cancer, 3 patients with tuberculosis requiring treatment and 9 other benign lesions. It is therefore reasonable to suggest that the CATS method can be considered for the resection of small pulmonary lesions, which require histological confirmation and cannot be accessed in a less invasive way.

## Supporting information

**S1 Data. The raw data table sheet is available as a supplement.**
(XLSX)

**S1 File. Additional table have been uploaded as a supplement.**
(DOCX)

## Acknowledgments

We would like to thank our friend and colleague, Mr. Zizi Zhou (Department of Cardiothoracic Surgery, Shenzhen University General Hospital, China), for his valuable comments and review of our study.

Preliminary results of this study were presented at the Annual Meeting of the European Association for Cardiothoracic Surgery (EACTS), Lisbon (October 5, 2019).

## Author Contributions

**Conceptualization:** Ioannis Karampinis, Nils Rathmann, Michael Kostrzewa, Steffen J. Diehl, Eric D. Roessner.

**Data curation:** Ioannis Karampinis, Nils Rathmann, Eric D. Roessner.

**Formal analysis:** Ioannis Karampinis, Nils Rathmann, Eric D. Roessner.

**Funding acquisition:** Steffen J. Diehl, Stefan O. Schoenberg.

**Investigation:** Ioannis Karampinis, Nils Rathmann, Michael Kostrzewa, Steffen J. Diehl, Eric D. Roessner.

**Methodology:** Ioannis Karampinis, Nils Rathmann, Michael Kostrzewa, Steffen J. Diehl, Eric D. Roessner.

**Project administration:** Michael Kostrzewa, Steffen J. Diehl, Stefan O. Schoenberg, Peter Hohenberger, Eric D. Roessner.

**Software:** Michael Kostrzewa.

**Supervision:** Stefan O. Schoenberg, Peter Hohenberger, Eric D. Roessner.

**Validation:** Steffen J. Diehl, Stefan O. Schoenberg, Peter Hohenberger, Eric D. Roessner.

**Visualization:** Steffen J. Diehl, Peter Hohenberger.

**Writing – original draft:** Ioannis Karampinis, Nils Rathmann.

**Writing – review & editing:** Steffen J. Diehl, Stefan O. Schoenberg, Peter Hohenberger, Eric D. Roessner.

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
