## [Decision Letter · Decision Letter 0]

3 Aug 2021

PONE-D-21-12920

Computer tomography guided thoracoscopic resection of small pulmonary nodules in the hybrid theatre.

PLOS ONE

Dear Dr. Roessner,

Thank you for submitting your manuscript to PLOS ONE. After careful consideration, we feel that it has merit but does not fully meet PLOS ONE’s publication criteria as it currently stands. Therefore, we invite you to submit a revised version of the manuscript that addresses the points raised during the review process.

Reviewers have raised some constructive comments. Your study provides a descriptive information about intraoperative CT-assisted thoracoscopic surgery for small pulmonary nodules. To strengthen your manuscript, it would be better to provide comparative analysis including others' reports.

We look forward to receiving your revised manuscript.

Kind regards,

Hyun-Sung Lee, M.D., Ph.D.

Academic Editor

PLOS ONE

Journal Requirements:

Reviewers' comments:

Reviewer's Responses to Questions

**Comments to the Author**

1. Is the manuscript technically sound, and do the data support the conclusions?

Reviewer #1: Partly

Reviewer #2: Yes

2. Has the statistical analysis been performed appropriately and rigorously? 

Reviewer #1: N/A

Reviewer #2: I Don't Know

3. Have the authors made all data underlying the findings in their manuscript fully available?

Reviewer #1: No

Reviewer #2: Yes

4. Is the manuscript presented in an intelligible fashion and written in standard English?

Reviewer #1: Yes

Reviewer #2: Yes

5. Review Comments to the Author

Reviewer #1: The authors reported the C-arm cone beam CT guided localization of pulmonary nodule with hookwire.

This paper doesn't seem new and very informative.

How is the depth of pulmonary nodules and how deep lesion is able to be localized.

Authors should clarify the number of patients (12 or 13) showed a benign disease.

And, the percentage of the benign disease seems to be a bit high, which means that unnecessary surgery had been done.

Table 1 and Figure 1 need to be summarized according to disease categories.

Reviewer #2: This is a single-institution, retrospective cohort study that examines 50 patients with a history of intraoperative computed tomography assisted thoracoscopic surgery (CATS) in 2014 and 2019 to analyze the routine use of a CT-aided thoracoscopic approach to small pulmonary nodules in the hybrid theatre and examine the influence of the histology on the treatment algorithm. You have demonstrated that CATS enabled successful resection in all cases with minimal morbidity and the histological diagnosis led to a treatment change in 42% of the patients. You have concluded that the hybrid-CATS technique provides good access to deeply located small pulmonary nodules and could be particularly valuable in the emerging setting of lung cancer screening.

Your study provides the descriptive information about CATS as the one of nodule localization methods for VATS wedge resection of small pulmonary nodules. Your study is interesting, but you need to provide more data to emphasize the feasibility of CATS and the alteration of further treatment plan after this procedure.

1. Please generate the CONSORT diagram about patient selection for this study.

2. It would be better to generate a table of patient characteristics of 50 patients including tumor location. Normally distributed continuous variables are reported as mean/standard deviation; non-normally distributed continuous variables are reported as median and interquartile range.

3. Please summarize table 1. It would be better to move all the detailed patient information to the supplementary table. Please describe the detailed histology of non-small cell lung cancer.

4. Are there any cases of intraoperative surgical extension from wedge resection to lobectomy and mediastinal lymph node dissection after intraoperative histology confirmation by frozen samples?

5. It would be better to illustrate the treatment change after CATS. Further treatment has not been demonstrated in detail.

6. To emphasize the feasibility of CATS, please generate a table of CATS-related complications in detail.

7. You can emphasize the merit of CATS by reviewing other localization tools of pulmonary nodules.

8. Several typo errors such as ration were found.

6. PLOS authors have the option to publish the peer review history of their article (what does this mean?). If published, this will include your full peer review and any attached files.

Reviewer #1: No

Reviewer #2: No

---

## [Author Response · Author response to Decision Letter 0]

10 Sep 2021

Dear Dr. Lee, 

thank you for the opportunity to revise our manuscript and resubmit it. Here you can find a point- to- point response to both yours and to the reviewers’ comments. 

Kind regards, 

Ioannis Karampinis, Nils Rathmann and Eric Roessner

Response to comments from the editorial office:

Point 2: Grant information has been updated to be identical in both the sections you pointed.

Point 3: The data availability statement has been updated accordingly. In order to meet the reviewers’ requests, we have added 2 additional tables, which has been uploaded as suppl. material. This way our complete datasets are publicly available either in the body of the manuscript, or as a supplement. 

Response to reviewers’ comments:

Dear colleagues, thank you for reviewing our work and for the points you raised. We hope the changes made will meet your expectations. 

Response to reviewer 1:

Point 1: Concerning the depth and the location of the nodules, we have added a table as a supplement, where you can find the information that you requested. 

Point 2: The maximum depth that you can reach with this technique is probably not a question that we can directly answer. The deepest nodule in this study was a 7 mm lesion lying 53 mm deep in the lung tissue. The path length for the marking wire was 64 mm. However, it is probably the vascular and bronchial anatomy that limits the resection of these nodules and less the absolute depth of the lesion itself, as you can usually manage to place a stapler around the lesion and resect it, if you manage to locate it securely (in rare cases you can run an additional CT scan to make sure that the tip of the wire is enclosed in your stapler).

Point 3: There were 12 patients with benign disease but 13 benign lesions (one patient had two suspicious lesions and both of them turned to be anthracotic lymphnodes, patient #14).

Point 4: Regarding the benign histology of the resected lesions, you are absolutely right. In this study we found a benign pathology in 25% of the cases, which is clearly high. However, the patient population of this study is different from the populations presented in all the well-known trials of patients with incidentally diagnosed solitary nodules or patients undergoing screening for lung cancer. The patients discussed here are not healthy individuals but patients with another known solid tumor and suspected pulmonary metastasis (46 the 50 cases). It is certainly a difficult decision to offer surgery to such a patient. From our point of view and taking the very low complication rate into account, it is rather positive to be able to relieve 25% of these patients from the diagnosis of metastatic cancer.

Point 5: You requested to summarize table 1 and figure 1 according to disease categories. Table 1 has been summarized. The bigger table from the first submission was kept as suppl. material. Figure 1 presents the time that was required for the procedure in each patient. We are not sure, what exactly you mean by summarizing this table based on the histology, as the histology does not really affect the time required to perform the procedure according to our opinion. If you wish to see another figure please explain in detail exactly what type of figure you expect. 

Response to reviewer 2:

Point 1: Thank you for the issue you raised. As you pointed out, this is a single institution, retrospective study and not an interim analysis of data that were collected parallel to a formal prospective trial. It is therefore not possible for us to include a CONSORT diagram because the necessary data on enrollment, eligibility etc. are not available.

Point 2: Another table has been created and submitted as suppl. material including the information that you requested. We have changed the medians with means.

Point 3: The changes you requested have been made accordingly and are included in the table in the suppl. material.

Pont 4: This is an excellent comment. It is our policy not to extend surgery in this specific subgroup of patients (patient with a malignant background undergoing operation for suspected metastasis) directly to an oncological resection for two different reasons: first, the result of the frozen section that we receive does not always allow us to securely differentiate metastatic disease from a new primary tumour and we often have to wait for the immunohistochemistry. Second, these patients have not received formal staging for lung cancer according to the national guidelines so we might end up performing a lobectomy in a patient with extensive disease, which is not justified. We prefer to complete the staging and then discuss with the patient the possibility of a completion lobectomy with SND, follow up or other types of treatment, depending on the histology and stage of the disease. Our policy is different for patient without previous malignant disease. 

Point 5: The treatment changes have been added in table 1

Point 6 + 7: The purpose of this article was not to describe CATS as a method but to discuss the histopathological findings of this specific patient cohort, that was operated using this method. This is the reason why we did not go into further detail in the CATS complications or discussed this method compared with other localization techniques. However, due to the fact that the latter was pointed out from both the reviewers and the editor we have added a paragraph in the discussion focused on this point.

Point 8: Thank you for pointing this out, we have cross-read the article again for typing errors and hope we have not missed further errors.

---

## [Editor Report · Decision Letter 1]

8 Oct 2021

Computer tomography guided thoracoscopic resection of small pulmonary nodules in the hybrid theatre.

PONE-D-21-12920R1

Dear Dr. Roessner,

We’re pleased to inform you that your manuscript has been judged scientifically suitable for publication and will be formally accepted for publication once it meets all outstanding technical requirements.

Kind regards,

Hyun-Sung Lee, M.D., Ph.D.

Academic Editor

PLOS ONE
---

## [Editor Report · Acceptance letter]

25 Oct 2021

PONE-D-21-12920R1 

Computer tomography guided thoracoscopic resection of small pulmonary nodules in the hybrid theatre. 

Dear Dr. Roessner:

I'm pleased to inform you that your manuscript has been deemed suitable for publication in PLOS ONE. Congratulations! Your manuscript is now with our production department. 

Kind regards, 

on behalf of

Dr. Hyun-Sung Lee 

Academic Editor

PLOS ONE